developmental information flow; plant biomechanics; plant morphogenesis; surface geometry; waveguides.

**Corresponding author:**
Email: philip.lintilhac@uvm.edu

**Associate Editor:**
Dr. Olivier Hamant

# Wilhelm Hofmeister revisited: Meristematic surfaces channel information flows in organogenesis

Philip M. Lintilhac

Department of Plant Biology, The University of Vermont, 05405, USA

## Abstract

Plant development relies not only on intracellular biochemical signals but also on physical information that is transmitted across cells and tissues. In growing plant organs, surface geometry and mechanics can act together to channel stress and strain signals beyond the single cell, effectively creating *trans-cellular* communication pathways that are robust, accurate and instantaneous. It follows that meristematic surfaces act as stress-mechanical *waveguides* to constrain and redirect internal stress fields. The orientation and patterning of these stress fields correlates with the placement of new cell walls during cell division, thereby linking surface geometry to tissue histogenesis. Here, I consider how meristem surfaces may contribute to developmental signaling via mechanical force transmission. I argue that surface curvature and tissue biomechanics can form a self-sustaining feedback loop: together, they shape force transmission trajectories, which in turn guide the fundamental decision-making processes that determine cell plate orientation during cytokinesis, thus altering organ shape.

## 1. Introduction

Every student of plant anatomy and structure has surely been dazzled by the beauty and regularity of plant materials seen in section. What can we learn from the patterns frozen into plant tissues about the logic of plant development? These questions have a long history in plant science. The study of tissue patterning and cell division orientation is one of the oldest topics in the study of basic plant structure and morphogenesis. The first and perhaps the most influential of these early studies were those of Wilhelm Hofmeister (1824–1877), who attempted to articulate basic rules governing the patterning of growing plant tissues. Even more remarkable was Hofmeister's ability to see past the flood of descriptive detail characteristic of early approaches to plant anatomy and begin to formulate the general principles that underlie plant development and morphogenesis. Hofmeister began to see plant development as the manifestation of basic physical behaviours that account for much of the repetitive patterning of plant tissues. Hofmeister's Rule (Hofmeister, 1863) was based on the observation that cell division orientations tended to be in a plane perpendicular to the long axis of the cell. It was one of the first attempts to codify an underlying logic to the process of plant histogenesis.

Later, the work of Kny (1896) and of Otto Schüepp (1917) proposed the presence of hidden force fields in meristematic structures, followed later by Snow and Snow (1951) who attempted to classify tissue stresses and strains using microsurgical methods. More recently, the biomechanics of tissue stresses and strains and their roles in the morphogenetic cycle of plant meristems have become a topic of increasing interest (Dumais & Steele, 2000; Echevin et al., 2019; Hamant et al., 2008; Sampathkumar, 2020). Considerable effort has been devoted to understanding the role of physical constraints on plant organogenesis and attempts to define the role of surfaces in plant organogenesis have been proposed (Burian et al., 2013; Kwiatkowska, 2004; Nemec-Venza et al., 2025).

The biology of surfaces can be studied in many different contexts, but it should be noted that there is a distinction to be made between an epidermal layer, which is a basic tissue type, comprising one or more cell layers of varying thickness, and the surface itself, which is

a boundary condition with no thickness. Epidermal surfaces play critical roles in development and morphogenesis by defining organ geometries and physically constraining underlying tissues (Kutschera & Niklas, 2007). Surfaces may also define the interfaces between disparate control systems, allowing us to think in terms of intracellular and extra-cellular information systems and the translation of one into the other at the plasma membrane. Here, I want to explore how surface-related mechanical cues serve as integral developmental signals in plants.

The central proposition that animates this discussion is that many critical processes in plant development are beyond the reach of cytoplasmic molecular signals and beg to be understood as *transcellular* signaling systems that can span multicellular regional domains, implying that physical behaviours originating in one region of a growing structure are deterministic, and can target and modify behaviours in another region, whereas signalling systems based on shifts in molecular populations, while highly specific at the stereochemical level, are fundamentally stochastic and tend to dissipate with time and distance (Lintilhac, 2022).

Mechanical signals propagate through hydraulically turgid and mechanically coupled tissues much faster than chemical diffusion or other forms of directed transport, effectively coupling distant regions of an organ instantaneously. Physical signals, like acoustic signals, can travel through mechanically coupled materials at the speed of sound, and under the right conditions, can be actively focused on distant locations in three-dimensional space. They are also capable of working in concert with the more familiar hormonal and transcriptionally driven networks to coordinate growth (Hamant et al., 2008; Heisler et al., 2010; Nakayama et al., 2012). Mechanical signals can remodel cytoskeletal anatomy at the level of the single cell (Hamant et al., 2019) and re-orient the division plane precisely with respect to the lines of maximal stress (Louveaux et al., 2016), thereby linking tissue-scale force transmission to transcriptionally mediated intracellular processes. In this regard, mechanical signaling may be seen as another layer of control to in plant morphogenesis, enhancing the robustness and spatial precision of developmental programming.

## 2. Apical meristems

Apical meristems may be among the most successful adaptations of plants to life on dry land, making it possible to generate long-lived, multicellular structures hundreds of meters tall, with lifespans measured in thousands of years. Anatomically, apical meristems are multicellular domes of turgid embryonic tissue covered by one or two tiered epidermal layers (Wegner, 2000). Growth in the surface layers is anisotropic, favouring expansion in the plane of the surface. Epidermal meristem cells typically support turgor pressures in the range of 1.0 MPa (Beauzamy et al., 2015), which makes it possible for expanding cells to generate similarly scaled compressive forces that radiate as directional force fields propagating from cell to cell through the surrounding tissues. Ultimately, it should be possible to decompose these force fields into orthogonal networks of tensile and compressive stress trajectories describing the mechanical stress conditions at any point in the growing structure.

In actively growing regions such as meristems, where apoplastic continuity provides tissue-wide mechanical coupling, increases in surface and volume result in distinct regions of 'locked-in' stress (Frocht, 1941; Heywood, 1969) where tensions and compressions dominate, resulting in the deformation of surface layers into ridges

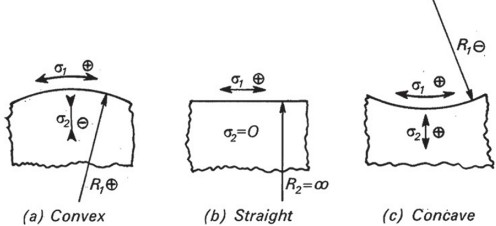

The two principal stresses at a free boundary are necessarily parallel and normal to the boundary, with the principal stress normal to the boundary being by definition of zero magnitude at the boundary.

- The principal stresses are of opposite sign when the boundary is convex.
- The principal stress normal to the boundary is zero locally if the boundary is straight.
- The principal stresses are of the same sign if the boundary is concave.

After Heywood, (1969)

**Figure 1.** Transverse stress behaviour at a free boundary.

and valleys which mature into leaf primordia and leaf axils, (Green et al., 1996; Hernández-Hernández et al., 2014).

Given that stress distributions are invisible to the outside observer, the assignment of stress directionalities in growing structures can be complex, but it can be approached experimentally using a variety of technologies, including photoelastic modelling (Heywood, 1969), which was widely used in the early 20th century. More recently, stress patterns have been determined using the finite element method (FEM) (Bozorg et al., 2014; Oliveri et al., 2019). Both methodologies allow for the mapping of force transmission trajectories in solid materials. In both cases, stress distributions must be separated into their tensile and compressive components, a necessity that is made more difficult by the fact that stress distributions are invisible to the outside observer. However, simple rules relating surface geometry to stress distribution have been used to decompose transmitted force fields into their component tensile and compressive families of stress (Figure 1).

These empirically derived rules have been used to identify and separate tensile and compressive stress regimes in mechanically loaded structures with curved surfaces (Heywood, 1969). When applied to a plant apical meristem, this means that, for a convex meristematic dome, the stresses running in the plane of the surface and those running in a radial direction (normal to the surface) will be of opposite sign; conversely, in concave regions, tangential and radial stresses will have the same sign. Therefore, compressive stresses generated by periclinal cell expansion tend to follow the contour of the surface layer, while tensile stress trajectories resulting from the apically directed bulge run normal to it.

Applying these rules to the anatomical reality of cell plate orientation and surface stress patterning in growing plant tissues, periclinal cell layers propagate by anticlinal divisions with new division walls being installed in a plane perpendicular to the plane of the surface. This means that cell expansion in the plane of the surface generates a surface compressive layer.

The surface of the meristem then becomes a landscape of convexities and concavities that drive the morphogenetic engine of the apical meristem, including axillary bud formation (Lintilhac & Vesecky, 1980). From a tissue-wide perspective, cell plate orientations at cytokinesis may thus be seen as a proxy for the directions of the primary principal stress acting through any cell at the time of cytokinesis (Höfler M. 2024) thereby providing a way to map the

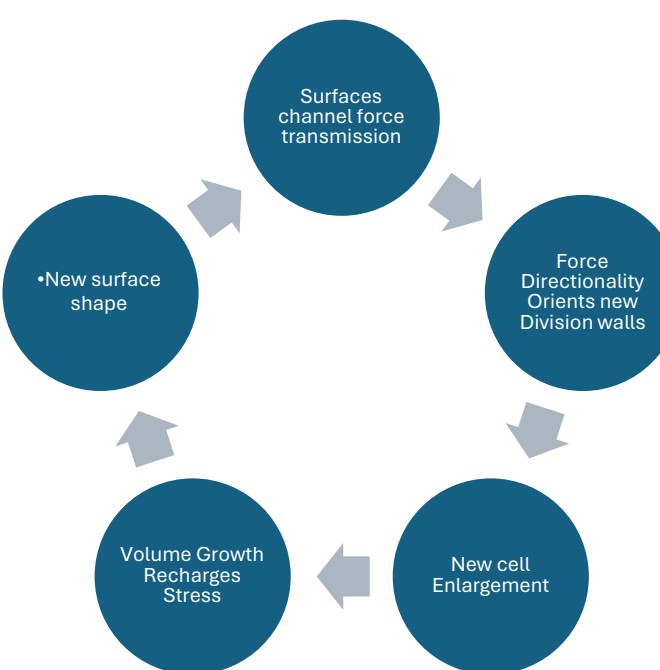

**Figure 2.** Force transmission and surface topography are linked in a feedback loop that drives morphogenesis.

flow of force transmission during primary growth. The patterning of plant tissues tells the story of tissue mechanics.

The interdependence of surface geometries, cell growth anisotropy and cell plate orientation tells us that plant meristems may be understood as shape-generating structural automata whose time-dependent behaviour is complementary to, but distinct from, the underlying transcription-based command structure that inhabits all living cells. Local surface shape *organizes* the emerging pattern of cell divisions. When we adopt the position that these self-sustaining feedback cycles represent the time-dependent trajectory of morphogenesis, developmental decisions relying on inputs from scripted, transcriptionally dependent information flows recede into the background, becoming less significant in the timing of events than the deterministic inertia of the meristem itself. Surface conformation, growth anisotropy and cell plate orientation appear to be locked in a feedback loop that forms the basis of morphogenetic time itself (Figure 2).

## 3. Mechanobiology of cell wall orientations

It is instructive to note the similarities between acoustic signals and mechanical signals. Mechanical energy, like acoustic energy, is transmitted as tensile and compressive force fields radiating through a material, and, like acoustic signals, mechanical energy is channelled and focused by the surfaces that constrain it. The forces generated by cell enlargement and tissue growth are also transmitted from cell to cell at the speed of sound, constrained by the outer surface but never propagating beyond it. In this way, the surfaces of growing plant organs may be considered to act as waveguides for the mechanical signals that are generated by plant meristems, just as surfaces can serve as waveguides for acoustic information transfer. Forces generated in one region radiate throughout a structure as families of orthogonal principal stresses, following surfaces that control their directionality.

In living plant materials, these families of orthogonal principal stress have often been shown to be reflected in the orientation of

plant cell walls, (Lintilhac & Vesecky, 1984) and particularly in the placement of newly formed cell plates at cytokinesis (Louveaux et al., 2016; Lynch & Lintilhac, 1997; Serra et al., 2024; Serra & Robinson, 2020). This leads to the proposal that the newly formed cell-plate finds its final orientation in a plane that is free from shear stress, necessarily placing it in a plane normal to the primary principal stress (Hernández-Hernández et al., 2014; Lintilhac, 1974). Ultimately, this means that the patterns of cell walls in any section of actively dividing plant material serve as a proxy for the stress directionalities that prevail at cytokinesis. How stress-mechanical signals are interpreted at the cellular level is a question of fundamental importance that will doubtless be resolved when we fully understand the capabilities of the cytoskeleton acting as a complex, multi-axial strain gauge.

At the cellular level, responses to mechanical stimuli lie largely in the functional domain of the cytoskeleton, which acts as a tuned tensegrity structure (Ingber, 2008). Similarly, cell plate orientations at cytokinesis are mediated by an elaborate assemblage of cytoskeletal elements and motor proteins that we call the pre-prophase band and phragmoplast, and are resolved by the coalescence of vesicle-bound cell wall precursors into the nascent cell plate; leading us to the conclusion that plant cells are well equipped to interpret and react with precision and speed to their stress-mechanical environments, optimizing cell wall patterning and tissue architecture accordingly.

## 4. Conclusions

Traditional developmental biology does not do justice to the central role that surfaces play in plant organogenesis. We often tend to assume that the ultimate control of development and morphogenesis lies in the transcription-mediated shifts in the molecular populations that drive intracellular information systems, but evolution has made use of diverse information systems to actualize the life-histories of plants. Multi-dimensional controls need to be understood in terms of multi-dimensional information systems.

While surface geometry is not something that can be isolated and purified in the sense that a molecule can, it can have profound effects on the surrounding cellular landscape and on the forces propagating through tissues. In the land plants, surface topography has evolved as an action element in the many developmental feedbacks that channel information flows and integrate the various levels of control. The underlying message is that the relationship between surface conformation and cell plate orientation is foundational in plant morphogenesis. Stress-mechanical signalling, focused and channelled by surface topography, has evolved into a robust and predictable information structure.

Physical force transmission as a source of developmental information is not restricted to the plant kingdom. It is well known that the architecture of bone and muscle is deeply dependent on physical inputs (Wolff, 1892), as is the structure of the eye and many of the essential features of early embryogenesis (Okuda et al., 2018) but in the plant kingdom, where cells share rigid cell walls that never shift position, and where tissues are permanently mechanically coupled, the cell wall patterning that we see in section may be read as a permanent record of the flows of mechanical signals through multicellular tissues.

Trans-cellular signalling in plants does not necessarily require secreted diffusible factors to carry information from one extracellular location to another. Instead, it can occur via the continuous elastic medium provided by the extensive mechanical

coupling of the apoplast and the ability of the cytoskeleton to act as a multi-axial strain-gauge. The surfaces of plant organs create invisible physical circuits for the transmission of mechanical information. The advantage of this form of information exchange is that force transmission in tissues is temperature-insensitive, instantaneous and directionally precise, qualities rarely seen in chemical signalling.

As research in plant biomechanics progresses, it is revealing an overarching layer of developmental control that is singularly adapted to the sessile, high-pressure and environmentally exposed growth habit of the land plants. In summary, plant surfaces have a life of their own in development: they constrain and channel the mechanical signals that orchestrate growth. Recognizing this reality opens new opportunities for a unified understanding of morphogenesis, where anatomy, physics and biology converge.

From a developmental point of view, plant apical meristems (shoot tips and root tips) can be regarded as physically integrated, surface-generating automata, establishing a self-sustaining biomechanical cycle whose components include cell plate orientation, cell enlargement, shape change, surface geometry, stress redistribution and cell plate orientation again (Lintilhac, 2014).

With the unfolding of plant biomechanics as a core discipline, plant biology may be entering a new era of theoretical integration and practical understanding. Surfaces are embedded in the circuitry controlling the generation of shape and form and serve more functions than just interfacing an organism with its environment. Wilhelm Hofmeister's intuitions are proving to be largely correct (Kaplan & Cooke, 1996). Plant tissue patterning and organogenesis can be approached in terms of a few relatively simple structural rules working together to create shape and form. But the rules only become apparent when the developmental context is correctly defined.

The confluence of molecular and physical control systems is uniquely accessible and transparent in the land plants, revealing the inherent logic of plant structure and development in a way that can never be achieved through molecular studies alone. The architectural nature of plant growth, expressed in the language of surfaces and volumes, holds the promise of leading us toward new experimental methodologies and a more comprehensive understanding of plant development

**Open peer review.** To view the open peer review materials for this article, please visit http://doi.org/10.1017/qpb.2026.10044.

**Competing interest.** The author declares none.

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
