## [Reviewer Report]

In this short review, the Author postulates the unique role of meristem surfaces in tissue (organ) mechanics and especially in instantaneous transcellular signalling regulating plant development and morphogenesis. The review is concise, logically constructed and most of all thought-inspiring. It is clearly related to Hofmeister’s contribution to plant science.

I have only few suggestions on how to improve the presentation:

(i) a schematic figure in which tensile and compressive stress trajectories were illustrated for various SAM shapes as well as empirical rules that govern stress behavior, would help to follow the reasoning in section “Apical meristems”

(ii) some statements are not quite so clear and in my opinion would need an explanation or correction. These are:

“symplastic structures” (page 2, 1 paragraph of Apical meristems section) – why only symplastic structures are referred to? Would it not be rather interlocked symplast and apoplast?

“tensions and compressions emerge as distinct regions of “locked-in” stress” (page 3, paragraph 2) – please explain in more detail;

“compressive stresses generated by periclinal cell expansion ….” (page 3, last paragraph) – please explain how this relates to or could be affected by growth (expansion) of inner cells?

(iii) finally there are some editorial corrections needed:

“physical behaviors in one region of a growing structure can target and modify behaviors in a distinct region” (page 2, paragraph 2) – is it not DISTANT regions rather than DISTINCT?

Please unify the format of citations in the text

I could not find a legend for figure 1?

Finally, in some places comma needs to be replaced by a dot.

---

## [Reviewer Report]

This paper presents the opinion that fields of mechanical stresses that build in meristematic surfaces during growth drive intra and inter-cellular processes and provide key information propagating fast in tissues. This information is assumed to be complementary to that of chemical pathways and to feedback on cell plates to synthesis according to the stress field main directions.

The idea that stresses forming in epithelial tissues are used during organogenesis to orient divisions or to feedback more generally on gene regulation has a long history, and various recent developments advocate for such a key role of mechanics in morphogenesis in both plants and animals. Such an opinion paper in this respect would be timely. However, despite the importance of the subject, I cannot recommend its publication as it stands, for the following main reasons:

- The paper seems to anchor an important part of its analysis in a book from the late sixties difficult to obtain, that was published in the domain of material science (photoelastic materials) and I believe not much known to the QPB community. Main assumptions and results from this approach should be presented and discussed at least. Limits of this approach should be also discussed as well as how this analysis is related to multicellular plant tissues.

- The presentation of stress patterns related to shape is oversimplified. Recent results show that when inner layers are taken into account, stress patterns are significantly modified with respect to the prediction of stresses in inflated shells with similar external geometries. Lots of recent work are contributing to the discussion of stress patterns in growing tissues and are not cited here (e.g. Nakayama 2012, Louveau 2016, Ali et al. 2019, Long et al 2020, Collet et al. 2025, etc.), although of clear relevance to the paper’s topic.

- I could not understand the rationale or the origin of the rules #1-4. These rules, their assumptions and their derivation should be clearly explained. It also seems almost impossible to understand non ambiguously without any figures.

- I am not convinced that these rules are even correct. I can’t see any general foundations for them in a multicellular turgid structure. This gives the impression that the paper overgeneralizes some observations and I don’t see general empirical or even theoretical support for these claims.

- In different places, the author implies that compression forces result from tissue growth. An inflated ellipsoid of revolution type of shell structure, shows circumferential compressive stresses in its outer region circumferential region, as soon as the ratio between its height to its radius is less than square root of 2. This is due to shape geometry only, and not necessarily to growth. This classical result in solid mechanics was already recognized in Dumais and Steele 2000 for example. Such an ellipsoid under sufficient pressure might have a geometry that induces only tensile circumferential stresses everywhere if the radius ratio of the inflated structure is more than square root of 2. However, deflating it it a bit, may change its geometry in such a way (flattening its dome rapidly and reaching a radius ratio < square root of 2) that compressive circumferential stresses start to appear on the outer part of the dome. Just by deflation. This shows that one could induce circumferential stresses without growth (only by deflation), and this is contradictory to what is claimed in rules #3 and 4 (as far as I understand).

- The paper is very partial in citing previous works, where similar ideas on the importance of stresses in regulating development have been proposed, in particular recent works. The bibliography contains 26 items, and only 11 cover the last 25 years, of which 3 are from the author. Which means that the reference to the vast recent literature work lies in 8 (i.e. less than one third) of the cited papers (note that some papers are note cited with the correct year, e.g. Dumais-Steele 2014, should be Dumais-Steele 2000.). For example, the work on FEM models is acknowledged as important in studying stress patterns in one sentence, but without providing any reference or any analysis for it.

- To my knowledge, no strict dependence between cell plate division and stress patterns have been ever reported in the shoot apical meristem for instance (contrary to what is claimed in the general sentence: "applying these rules to anatomical reality of growing plant tissues we note the strict dependence of cell plat orientation on surface stress patterns.)

- Stress patterns in meristems are actually not known precisely. Various groups worldwide are at the moment working on mapping stresses using different measurement techniques (AFM, inverse modeling, pressure probes). So I don’t think that we can yet assert so strongly that stresses are channeling morphogenetic information in meristems based on clear experimental data (although this is a very attractive assumption for the research community at the moment).

---

## [Reviewer Report]

I have been the reviewer of initial submission of this manuscript and thus now I refer only to how the manuscript has been revised. All the reviewer’s suggestions were addressed by the Author and there are only a few mainly editing mistakes that need to be corrected.

1. Please correct the citations of Hernandez-Hernandez et al. 2014 - it is referred to as “Hernandes & Hernandes 2014” in the text

2. Please check the reference to Figures: on page 5 (the last line) “Table 1, Figure Y” is referred to where it should be Figure 1; on page 8 (the second line) Figure 1 is referred to where it should be Figure 2

3. on page 12, the last paragraph of the main text – please consider whether “instincts” is the best term. Would “intuitions” not be better?

4. Last but not the least – the legend of Figure 1 is confusing for me. Please explain the schemes more straightforward, especially what direction of stress is referred to. For example, it is confusing to use the term normal stress (as different from shear stress, yes?) in the same sentence as perpendicular direction, because the normal direction is the same as perpendicular … Also please explain which “principal stress being by definition of zero magnitude” is referred to in the first sentence.

---

## [Reviewer Report]

Thank you for this contribution.

It is brilliant to get such a well-written manuscript from such a trail-blazing leader in the field.

The current manuscript will whet reader’s appetite for thinking about physics in plant biology and considering engineering concepts. I am confident it will achieve that goal.

There are several questions that arise from such considerations and I for one would be very keen to hear the author’s views on big questions in the field, puzzling observations, or recommendations for more junior scientists for what might be fruitful areas to work on or how upcoming technologies may help drive the field forward.

For instance, all surfaces behave similarly in terms of transmitting mechanical changes - both biological and non-biological. What distinguishes these? What are the critical questions? How might mechanical cues be processed and remembered (which would be necessary given their speed relative to biological regulatory changes)?

These are just examples - and I am sure the author has his own ideas and views that the community could massively benefit from -. but I think such additions would offer a route into lifting a taster opinion piece into a vision and start of a roadmap for the field.